# Elevated Blood Pressure Occurs without Endothelial Dysfunction in a Rat Model of Pulmonary Emphysema

**DOI:** 10.3390/ijms241612609

**Published:** 2023-08-09

**Authors:** Elodie Desplanche, Pierre-Edouard Grillet, Quentin Wynands, Patrice Bideaux, Laurie Alburquerque, Azzouz Charrabi, Arnaud Bourdin, Olivier Cazorla, Fares Gouzi, Anne Virsolvy

**Affiliations:** 1PhyMedExp, Université de Montpellier, INSERM, CNRS, 34295 Montpellier, France; elodie.desplanche@inserm.fr (E.D.); quentin.wynands@gmail.com (Q.W.); patrice.bideaux@inserm.fr (P.B.); laurie.alburquerque@inserm.fr (L.A.); azzouz.charrabi@inserm.fr (A.C.); olivier.cazorla@inserm.fr (O.C.); 2PhyMedExp, Université de Montpellier, INSERM, CNRS, CHU de Montpellier, 34295 Montpellier, France; pe-grillet@chu-montpellier.fr (P.-E.G.); a-bourdin@chu-montpellier.fr (A.B.); f-gouzi@chu-montpellier.fr (F.G.)

**Keywords:** endothelial dysfunction, blood pressure, heart failure with preserved ejection fraction, chronic obstructive pulmonary disease, emphysema

## Abstract

Chronic obstructive pulmonary disease (COPD) is an inflammatory lung disease involving airway closure and parenchyma destruction (emphysema). Cardiovascular diseases are the main causes of morbi-mortality in COPD and, in particular, hypertension and heart failure with preserved ejection fraction (HFpEF). However, no mechanistic link has currently been established between the onset of COPD, elevated blood pressure (BP) and systemic vascular impairment (endothelial dysfunction). Thus, we aimed to characterize BP and vascular function and remodeling in a rat model of exacerbated emphysema focusing on the role of sympathetic hyperactivity. Emphysema was induced in male Wistar rats by four weekly pulmonary instillations of elastase (4UI) and exacerbation by a single dose of lipopolysaccharides (LPS). Five weeks following the last instillation, in vivo and ex vivo cardiac and vascular functions were investigated. Exacerbated emphysema induced cardiac dysfunction (HFpEF) and a BP increase in this COPD model. We observed vasomotor changes and hypotrophic remodeling of the aorta without endothelial dysfunction. Indeed, changes in contractile and vasorelaxant properties, though endothelium-dependent, were pro-relaxant and NO-independent. A β1-receptor antagonist (bisoprolol) prevented HFpEF and vascular adaptations, while the effect on BP increase was partial. Endothelial dysfunction would not trigger hypertension and HFpEF in COPD. Vascular changes appeared as an adaptation to the increased BP. The preventing effect of bisoprolol revealed a pivotal role of sympathetic hyperactivation in BP elevation. The mechanistic link between HFpEF, cardiac sympathetic activation and BP deserves further studies in this exacerbated-emphysema model, as well as in COPD patients.

## 1. Introduction

Chronic obstructive pulmonary disease (COPD) is an inflammatory lung disease characterized by airflow obstruction and/or emphysema. Besides the pulmonary impairment, COPD patients experiment a high burden of cardiovascular comorbidities. Meta-analyses have shown a two to five times increase in the prevalence of cardiovascular diseases in COPD [1,2]. Of these, high blood pressure (BP) is the most common, being present in approximately 50% of COPD patients [3,4] and having a 33% increased mortality risk [1]. Vascular remodeling and dysfunction have been observed in COPD patients, including in particular peripheral endothelial dysfunction [5,6] and also increased arterial stiffness [7,8], arterial wall thickening [9], enlargement of the conductive arteries [10] and capillary rarefaction in the skeletal muscle [11,12]. Endothelial dysfunction, often described as a common feature of COPD, is defined as a state of imbalance between endothelium-derived relaxing and constrictive factors (generally related to an impaired NO bioavailability) that favors vasoconstriction [13]. A significantly lower endothelium-dependent vasodilatation was observed in COPD patients when compared with non-COPD controls [12,14], and this was even more pronounced in patients with COPD and cardiovascular disease [15].

While a better description of vascular impairment in COPD patients has led to an increased awareness of cardiovascular comorbidities [16], there is still no vascular therapy that can be specifically recommended or tailored for COPD patients, because no mechanistic connection has ever been established between the onset of COPD, elevated BP and vascular remodeling and dysfunction [16]. Observational clinical studies do not provide evidence for a causal relationship between COPD, vascular impairment and hypertension and cannot isolate potential underlying mechanisms [17]. Thus, the vascular impairment in COPD may either be the cause, through impaired vasorelaxation, or the consequence, through arterial thickening and enlargement, of elevated BP. Conversely, animal models of COPD are valuable tools for deciphering the mechanisms of cardiovascular comorbidities in COPD [18,19]. In particular, the elastase (ELA) and lipopolysaccharide (LPS)-induced model of exacerbated emphysema combining weekly instillations of a proteolytic enzyme elastase with the concomitant use of LPS-endotoxin is a validated model with systemic impairment [20]. Recently, we confirmed that ELA-LPS rats exhibit a pulmonary impairment mimicking COPD and emphysema [21]. Thus, describing vascular remodeling in this ELA-LPS rat model could allow for understanding the causal link leading to BP increase and vascular impairment in COPD. In addition, vascular impairment in COPD has been attributed to several factors, including hypoxia, autologous nervous system, inflammation, oxidative stress and aging [22]. However, the potential contribution of sympathetic hyperactivity to those vascular impairments has not been explored thus far. This role could be assessed using this ELA-LPS rat model since sympathetic hyperactivity impacting the cardiac system (diastolic dysfunction and tachycardia) has been previously described [21]. In addition, a cardioselective β1-receptor antagonist was able to prevent the onset of diastolic dysfunction. Bisoprolol treatment seemed also appropriate to study the link between sympathetic hyperactivity, hypertension and endothelial and vascular dysfunction.

The aim of the present study was to assess systemic blood function in a model of exacerbated emphysema using ELA-LPS rats and to characterize endothelial or arterial functions and remodeling. Then, using the cardioselective beta-blocker (BB) bisoprolol, we aimed to evaluate the effects of sympathetic hyperactivity on the possible vascular changes evidenced in this ELA-LPS rat model.

## 2. Results

### 2.1. Cardiac Function and Maximal Exercise Oxygen Uptake (V′O_2_ Max)

At the cardiac level, left ventricle ejection fraction (LVEF) and cardiac output were not different between the three groups (Table 1). A significant increase in the left ventricle E/e′ index was observed in ELA-LPS when compared to Ctrl (*p* = 0.04), consistent with a diastolic dysfunction of the LV. The diastolic function in ELA-LPS rats was prevented by treatment with bisoprolol, as shown by the normal E/e′ index compared with Ctrl (*p* = 0.78) and lower index compared with ELA-LPS (*p* = 0.04).

Maximal exercise oxygen uptake (V′O_2_ max) was significantly lower in ELA-LPS (*p* < 0.001), with a reduction of 9% when compared to Ctrl (Table 1). This reduced maximal exercise capacity in ELA-LPS was not observed with bisoprolol treatment in ELA-LPS (ELA-LPSBB group) (*p* = 0.306). Moreover, no difference was also observed with ELA-LPS (*p* = 0.117). Thus, we observed that bisoprolol treatment prevented exercise intolerance in ELA-LPS rats.

### 2.2. Blood Pressure

As depicted in Figure 1a,b, respectively, systolic and diastolic blood pressures were both increased in ELA-LPS (*p* < 0.0001 vs. Ctrl), logically leading to an increase in mean arterial pressure (120.6 ± 1.7 mmHg) compared to Ctrl (102.3 ± 1.5 mmHg, *p* < 0.0001). In the ELA-LPS-BB group, systolic and diastolic BPs were significantly lower than in ELA-LPS group (*p* = 0.045 and *p* = 0.044, respectively). Furthermore, in ELA-LPS-BB, even though BP was reduced, it was not normalized. Indeed, diastolic, systolic and mean (113.9 ± 2.5 mmHg) arterial pressures were still elevated when compared to Ctrl (*p* = 0.003, *p* < 0.0001, *p* = 0.0005, respectively). Pulse pressure was not different between groups (*p* = 0.139) (Figure 1c). It should be noted that HR determined during tail-cuff BP measurements did not differ between the three groups (*p* = 0.305) (Figure 1d).

### 2.3. Morphological Remodeling of Rat Aorta

We next explored the structural remodeling of the aorta by evaluating the arterial wall morphology and elastin content by histological staining of sections of the aorta. Morphometric measurements revealed changes in the vessel walls (Figure 2). Media thickness was decreased by 8% in ELA-LPS (*p* = 0.046) and normal in ELA-LPS-BB (*p* = 0.617) when compared to Ctrl (Figure 2b). The internal lumen diameter was decreased in ELA-LPS-BB by 17% when compared to Ctrl (*p* < 0.0001) and by 20% when compared to ELA-LPS (*p* < 0.0001) (Figure 2c). These changes reflected hypotrophic remodeling in ELA-LPS with thinning of the arterial wall. Hypotrophy was attenuated in ELA-LPS-BB in which outward eutrophic remodeling was observed in addition. The percentage of elastin fiber was not different between the three groups (*p* = 0.163) (Figure 2d).

### 2.4. Evaluation of Vascular Contractile Properties of Aorta

The contractility of aortic rings was evaluated with and without endothelium using the depolarizing agent KCl, the α1-adrenergic agonist Phe and the thromboxane A2 agonist U46619 (Figure 3). The agonists induced concentration-dependent contractions of the aorta either in the presence or in the absence of endothelium. In the presence of endothelium, the maximal contractile responses to KCl and Phe were reduced by 16% (*p* < 0.001) and 19% (*p* = 0.039), respectively, in ELA-LPS when compared to Ctrl (Figure 3c,d), while they were not altered in absence of endothelium in this group. In ELA-LPS-BB, the decreased contractility in the presence of endothelium was fully prevented for Phe (*p* = 0.01 vs. ELA-LPS), while it was still observed for KCl. No difference between the groups was observed with U46619 either in the presence or in the absence of endothelium (Figure 3e). The aorta sensitivities to KCl and Phe were not modified between Ctrl and ELA-LPS, as reflected by identical EC_50_ values (Table 2). On the other hand, we observed a decreased EC_50_ value for Phe in ELA-LPS-BB compared to ELA- which is not consistent with an increased sensitivity to Phe and might reveal adrenergic activation in ELA-LPS.

### 2.5. Evaluation of Vasorelaxant Capacity of Aorta

The vasorelaxant properties of the aorta were evaluated with Ach and SNP (Figure 4). Ach was used to assess the endothelium-dependent relaxant functionality of the vessel. With SNP, we evaluated the sensitivity of the aortic muscle to NO. Ach induced a concentration-dependent relaxation that was increased in ELA-LPS (Figure 4a) as revealed by a higher E_max_ value when compared to Ctrl (Figure 4c). The maximal Ach-induced relaxation in ELA-LPS was 12% higher when compared to Ctrl (*p* = 0.03). BB treatment partially modified Ach relaxation. Indeed, the E_max_ value was identical to that of both Ctrl (*p* = 0.94) and ELA-LPS (*p* = 0.2), suggesting an effect of BB treatment with a slight tendency to decrease Ach-induced relaxation otherwise increased in ELA-LPS.

SNP also induced concentration-dependent relaxation in all groups (Figure 4b). The sensitivity and maximal response of aortic smooth muscle to the NO donor were not different in all groups (Figure 4d). Only Ach responses were different between groups, suggesting that the modifications of vasorelaxation observed in ELA-LPS are endothelium-dependent.

The endothelium modulates vascular function through different regulation pathways involving mainly the release of vasorelaxant (NO, prostaglandin/prostacyclin) and vasoconstrictive (thromboxane, endothelin) substances. To investigate the involvement of these different mechanisms in the modifications observed in ELA-LPS, we studied the effects of various inhibitors on Ach vasorelaxant responses. Thus, concentration responses for Ach were measured in the presence of L-NAME and indomethacin, inhibitors of NO-synthase and cyclooxygenases (COX) enzymes, respectively (Figure 5). In Ctrl (Figure 5a) and in the presence of L-NAME, Ach-induced relaxation is almost completely suppressed, illustrating that NO release was the main regulatory pathway involved in the Ach response of the rat aorta. On the contrary, in the presence of indomethacin, Ach-induced relaxation is potentiated in Ctrl, certainly as a consequence of a reduced production of the vasoconstrictive prostaglandins consecutive to COX inhibition. The inhibitory effect of L-Name on Ach relaxation was still observed in ELA-LPS and in ELA-LPS-BB (Figure 5b,c). The analysis of the difference in the areas under the relaxation response curve of Ach in the presence and in the absence of L-NAME (delta AUC) showed no difference between groups, reflecting that the effect of L-NAME was not different in all groups (*p* = 0.36) (Figure 5d). Additionally, increased relaxation was still observed in ELA-LPS whether with or without L-Name (Figure 5e), which shows that NO pathway was not involved in that increase.

In addition, the potentiating effect of indomethacin on Ach-induced relaxation was not present in ELA-LPS (Figure 5b), while it was still observed in ELA-LPS-BB (Figure 5c). The analysis of the delta AUC for indomethacin showed differences between ELA-LPS and Ctrl (*p* = 0.002) and ELA-LPS-BB (*p* = 0.033), reflecting that the indomethacin effect differed between groups. Additionally, in the presence of indomethacin, Ach-induced maximal relaxation was identical in all groups (Figure 5e). Indomethacin suppresses the difference between groups for Ach-induced relaxation.

In summary, we observed that COX inhibition suppressed the difference between groups while NO synthase inhibition did not change it. This reflected the involvement of the signaling pathway mediated by COX activity in the increased relaxation observed in ELA-LPS.

## 3. Discussion

In the exacerbated elastase-induced emphysema rat model, mimicking COPD, we observed an increased blood pressure associated with an arterial hypotrophic remodeling without endothelial dysfunction. Systemic vascular functional changes included instead decreased contractile responses to agonists and increased relaxant responses to antagonists. In this COPD-emphysema rat model, the prevention cardioselective beta blocker bisoprolol induced partial improvements in BP and vascular function.

This ELA-LPS rat model is one of the most attractive for studying the systemic consequences of lung damage induced by pulmonary emphysema coupled with LPS-induced pulmonary inflammation. Pulmonary changes reported in animals exposed to elastase and LPS mimic features reported in mild COPD patients [23]. Moreover, as we previously demonstrated that a diastolic dysfunction that recapitulates a heart failure with preserved ejection fraction (HFpEF) was associated with lung emphysema in this model [21], it provides an opportunity to further study the cardiovascular mechanisms involved in COPD patients [18]. In the present study, the presence of HFpEF was confirmed in ELA-LPS by an elevated E/e′ ratio without modification of the left ventricular ejection fraction [24,25]. As expected, increased BP was also observed in these animals. Measurement using a validated tail-cuff device showed proportional increases in systolic, diastolic and mean arterial pressure with no variation in HR. The moderate BP increase observed in ELA-LPS (+18 mmHg for MAP) is consistent with mild hypertension or prehypertension observed in usual hypertensive models such as young hypertensive SHR rats. Additionally, the classical markers of COPD severity, i.e., body weight and exercise capacity reduction [26], did not change much in ELA-LPS animals. No variation in body weight was observed between groups, and the maximum oxygen uptake during exercise (VO’_2_ max)—an integrated assessment of the pulmonary, cardiovascular and muscle capacities—was only slightly reduced (9%).

Our data evidenced hypotrophic remodeling of the ELA-LPS aortic wall. This observation is original in the context of COPD, given that COPD patients usually showed aorta enlargement [10] and an increased risk of arterial wall thickening [9]. This hypotrophic arterial remodeling in ELA-LPS was not the consequence of systemic diffusion of porcine elastase, as elastase perfusion induced aortic aneurysms characterized by a progressive increase in the aortic diameter, a transient thickening of the arterial wall and an increase in elastic fibers [27,28]. In our ELA-LPS rats, no degradation of elastin fibers was detectable in the arterial wall. Similarly, LPS intratracheal instillations in mice were not able to induce such hypotrophic remodeling in pulmonary arteries [29]. Conversely, hypotrophic remodeling of conduit arteries has been reported in several experimental models associated with increased BP, like in ouabain-induced hypertensive rats [30], in late-pregnancy rats [31] and in genetically hypertensive rats [32]. Altogether, while the hypotrophy remodeling reported in our experiments was consecutive to the onset of the exacerbated-emphysema condition, this vascular remodeling could be mediated by an effect of BP increase on the vascular wall.

In ELA-LPS, contraction responses were decreased while relaxation responses were increased. These changes in vascular reactivity were clearly endothelium-dependent because the decreased contraction was not observed in the absence of endothelium. Moreover, the vasorelaxant responses differed with Ach and not with SNP. Additionally, inhibitor experiments suggested that this increased relaxation in ELA-LPS was not dependent on the NO-synthase, but rather due to increased activation of the COX-mediated pathway. Even though the vascular reactivity changes in ELA-LPS rats were endothelium-dependent, they are not in line with the definition of endothelial dysfunction, because the NO-dependent vasorelaxation was not impaired and the vascular reactivity imbalance was in favor of vascular dilation. Thus, our results may be discrepant with impaired endothelial function reported in COPD patients [12]. However, this impairment was not systematic in patients. Furthermore, it is not supported by experimental evidence in a COPD animal model.

Our study questions the link between the increase in BP and changes in vascular reactivity. Interestingly, the selective activation of the endothelium-dependent vasorelaxation pathway (eNOS activator, [33]) or inhibition in vasoconstriction pathways [34,35] induced a decrease in BP. Thus, it is unlikely that the decreased contraction and increased relaxation found in our ELA-LPS rats caused the BP increase. Conversely, this adaptation of the vascular reactivity could rather be the consequence of the BP increase in our ELA-LPS rats, as previously reported in genetically induced hypertensive rats [36]. In addition, before the onset of hypertension, borderline hypertensive rats showed an increase in acetylcholine-induced vasorelaxation and a decrease in phenylephrine-dependent contraction [37] that were not observed in SHR [38]. Such an adaptative mechanism of vascular reactivity to BP stressors has also been reported in the context of adrenergic hyperactivity. Indeed, in metabolic syndrome rats, increased vasopressive epinephrine in the plasma was compensated by increased eNOS vasorelaxation activity and a decrease in the phenylephrine-induced contraction, leading to a lack of BP increase [39]. Altogether, the changes observed in arterial function in ELA-LPS could constitute an adaptation to elevated BP rather than a primary mechanism, in line with the hypotrophic remodeling reported in these ELA-LPS rats.

We have previously demonstrated the beneficial effect of the cardioselective β1-receptor antagonist bisoprolol in preventing the onset of HFpEF [21]. Besides the HFpEF correction, the BP increase was partially prevented in ELA-LPS-BB. The reduction in BP in ELA-LPS-BB cannot be related to a decreased HR. Indeed, while previous experiments using telemetric HR assessment showed lower HR in ELA-LPS-BB rats (−45 bpm, −14%) and unchanged HR in ELA-LPS rats (+10 bpm, +3%), HR did not significantly differ between non-treated and treated ELA-LPS rats during tail-cuff BP measurements (354 ± 5 vs. 357 ± 15 bpm; *p*= 0.427). Also, no improvement of cardiac output after treatment can explain the reduction in BP in ELA-LPS-BB (through lower activation of the renin–angiotensin–aldosterone system). Last, the reduction in BP in ELA-LPS-BB cannot be related to a vascular effect. Indeed, bisoprolol is a highly selective β_1_-adrenoreceptor antagonist [40] with no sympathomimetic activity and vasodilator action [41], and no effect of the drug was reported on the vascular resistance of systemic arteries [42] and in hypertensive [43,44] or normotensive patients [45].

The effect of bisoprolol on BP could be caused by its effect on sympathetic nervous system activation in ELA-LPS-BB rats. Such an effect has been well demonstrated in hypertension [46], and sympathetic hyperactivity is involved in the pathogenesis of primary hypertension [47,48]. An adrenergic activation has been found in the heart of ELA-LPS rats (tachycardia and activation of the PKA-dependent pathway) and was prevented by bisoprolol treatment. In addition, β-blockers are sympatholytic, inhibiting the action of sympathetic neurotransmitters and modulating sympathetic neurotransmitter release [49]. Thus, a decreased release of catecholamines (epinephrine and norepinephrine) in ELA-LPS-BB rats could explain the partial prevention of BP increase in ELA-LPS rats. This effect could impact both the cardiac and the vascular system. Yet, the prevention of HFpEF in ELA-LPS-BB rats could have contributed to the BP effect because the HFpEF heart has appeared as a source of catecholamine in patients [50]. Conversely, the prevention of BP increase in ELA-LPS-BB rats could have prevented the HFpEF through a decrease in the postcharge of the heart. Thus, the sympathetic hyperactivity known to be involved in the pathogenesis of hypertension could play a role in the pathophysiology of COPD, through an “inflammatory reflex” [51].

Given our hypothesis of a secondary adaptation of the vascular remodeling and reactivity of the aorta to the increased BP, it was logical to observe the prevention of the functional and morphological remodeling of the aorta in the ELA-LPS-BB. Indeed, the endothelium-dependent vasorelaxation was corrected and the aortic contraction was normalized in ELA-LPS-BB. However, the prevention of the BP increase in ELA-LPS-BB rats was partial, meaning that the BP increase in ELA-LPS rats was not exclusively dependent on sympathetic activation. Other vasopressive mechanisms—like reactivity and remodeling of the distal resistance arteries—could play a larger role in BP increase in our COPD-emphysema model and should be addressed in further studies.

Some limitations of this study would be first that the animals were not exposed to cigarette smoke, which is a major, although not systematic, factor in COPD. However, no changes in vascular reactivity of the aorta were previously reported in a guinea pig model of cigarette-smoke-induced emphysema [35], suggesting that our findings are not irrelevant in the context of COPD. Second, our animal model is a mild-severity model of COPD, as revealed by the mild cardiovascular impairments. This could limit the assessment of pathological mechanisms involved in vascular impairment in COPD. However, it provides an opportunity to investigate the early cardiovascular mechanisms that may occur in COPD patients. Therefore, the relatively low severity of the model could explain the disparity between our findings and the published data for humans. Our observed increase in endothelium-dependent vasorelaxation may appear contradictory to previous results of meta-analyses of COPD, which have shown a significant decrease in endothelial-dependent vasorelaxation [5,6,14]. However, studies assessing invasive (acetylcholine-induced) or non-invasive (flow-mediated) vasorelaxation in COPD patients have not consistently demonstrated its impairment [52,53]. Moreover, using pulse-arterial tonometry, an impaired post-occlusive (endothelium-dependent) vasorelaxation was found in only 50 to 57% of patients [54,55,56], indicating that normal or even increased endothelium-dependent vasorelaxation may occur in at least 40% of COPD patients, in those with less severe pulmonary impairment [54,55,57]. Notably, some COPD patients with preserved exercise capacity and elevated BP have shown increased post-occlusive vasorelaxation [55], similar to our COPD-emphysema animal model. Overall, our results challenge the notion that endothelial dysfunction, traditionally considered a preclinical vascular impairment in COPD, is a critical step leading to hypertension and cardiovascular comorbidities in COPD patients [58,59,60].

## 4. Materials and Methods

### 4.1. Animal Ethics

All experimental procedures were conducted in accordance with the European Union Laboratory Animal Care Rules (2010/63/EU Directive) and NIH Guidelines. The project (APAFIS #13133) was approved by the local committee for Animal Care of Montpellier-Languedoc-Roussillon (No. CEEA-LR-9808) and the French Ministry of Research. Healthy male Wistar rats (seven weeks old) were maintained in our animal facility one week before experiments with free access to food and water.

### 4.2. Animal Model and Tissue Collection

Three groups were studied: control (Ctrl), animals with pulmonary exacerbated emphysema (ELA-LPS) and animals with pulmonary exacerbated emphysema treated with a beta-blocker (ELA-LPS-BB) as previously reported [21]. Experimental emphysema was induced under anesthesia (1% isoflurane) by intratracheal instillation of pancreatic porcine elastase (ELA, 4UI in 200 µL physiological serum/week, Sigma-Aldrich, Molsheim, France) for four weeks. To mimic COPD exacerbations related to recurrent pulmonary infections, rats received a single dose of LPS (2.5 mg/kg, Sigma-Aldrich, France) with the last ELA instillation. Some animals were treated with a β1-receptor antagonist (bisoprolol). Treatment started 24 h after the last ELA-LPS instillation and was administered in the drinking water at a concentration corresponding to 2.5 mg/kg/day (bisoprolol hemifumarate, MedChemExpress, Monmouth Junction, NJ, USA). Control animals receiving no instillation and no treatment were handled in the same conditions. Animals were investigated 5 weeks following the last pulmonary instillation. At the end of the protocol, physiological parameters were assessed by echocardiography, V′O_2_ max determination and blood pressure measurement. Then animals were euthanized by lethal injection of pentobarbital overdose. The aorta and blood samples were collected for ex vivo vascular reactivity and in vitro assays, respectively. After removal, the aorta was immediately immersed in a physiological saline solution (PSS, containing in mM 119 NaCl, 4.7 KCl, 1.2 MgSO_4_, 1.2 KH_2_PO_4_, 11 glucose, 25 NaHCO_3_, 2.5 CaCl_2_; pH = 7.4). Aortic tissue was cleaned from fat and connective tissues, cut into small rings and processed for the various experiments. Blood samples were centrifuged for 10 min at 1500× *g* and 4 °C, and plasma was frozen until use.

### 4.3. Maximal Treadmill Exercise Test with Measurement of Maximal Oxygen Uptake (V′O_2_ Max)

Maximal oxygen consumption (V′O_2_ max) was measured during an incremental exercise test on a metabolic treadmill coupled with a gas analysis system (Oxymax, Columbus Instruments, Columbus, OH, USA) as previously reported [21]. Animals fasted 6 h before the test. Rats were initially familiarized with running on a treadmill during 5 days, one week before the test [61]. On the day of measurement, after a 5 min period of acclimation, animals were subjected to incremental-speed exercise starting from 10 m/min with a gradual increase in steps of 5 m/min every 3 min until animal exhaustion. Exhaustion was defined as the animal remaining on the electrical stimulation grid without attempting to re-engage the treadmill within 15 s. The V′O_2_ and V’CO_2_ flow rates were calculated based on the measurement of the fractions in O_2_ (FO_2_, in %) and in CO_2_ (FCO_2_, in %), every 30 s, at the inlet and at the outlet of a sealed chamber. V′O_2_ max was defined as the higher value of V′O_2_ obtained (average of 30 s) before the stop.

### 4.4. Cardiac Function

High-resolution echocardiography was performed on a Vevo 3100 system equipped with a 20 MHz ultrasound probe (Fujifilm VisualSonics, Toronto, ON, Canada). Data were acquired under anesthesia (2–3% isoflurane inhalation) and monitoring of body temperature, ECG and respiration as previously described [62]. Morphological and functional cardiac parameters were characterized in the M-mode and B-mode from a short-axis view. Tissue Doppler imaging was performed to assess the early diastolic myocardial relaxation velocity wave (e′). Peak early (E) and late atrial contraction (A) mitral inflow waves were measured. Evaluated parameters included left ventricular ejection fraction (LVEF%) and E/e′ ratio (as an index of the left ventricle LV filling pressure). Offline image analyses were performed using dedicated Visual Sonics Vevo 3100 3.1.0 software.

### 4.5. Blood Pressure Determination

The CODA-6 tail-cuff system (Kent Scientific Corporation, Torrington, CT, USA) uses volume-pressure recording (VPR) to measure BP by determining tail blood volume. This provides measurements for hemodynamic parameters including systolic blood pressure (SBP), diastolic blood pressure (DBP), mean arterial pressure (MAP) and heart rate (HR). On day one, measurements for SBP, DBP, MAP and HR in both groups were recorded using a computerized CODA BP monitor, an occlusion tail-cuff and a VPR sensor (Kent Scientific, Torrington, CT, USA), according to the manufacturer’s instructions. Briefly, the CODA monitor was set to take measurements at 10 s intervals with a 15 s cuff deflation time, and an inbuilt test procedure for the condition of the occlusion cuff and VPR sensor was run prior to use each day. Animals were placed at 37 °C for 20 min before the recording.

### 4.6. Aortic Diameter and Media Thickness

For each animal, an aortic segment of 3 mm length was taken between the diaphragm and the hepatic artery, fixed in 4% PFA, embedded in paraffin, and cut into 6 µm serial cross-sections with a microtome. Sections were stained with Verhoeff’s solution, the first step in Van Gieson staining which highlights elastic laminae. Optical bright-field images acquired with a slide scanner (Nanozoomer, Hamamatsu, Japan) were digitally analyzed with simple image software (Fiji-ImageJ V1.8.0_172) (ImageJ, NIH, Bethesda, MD, USA) for morphometric analysis and elastin quantification. The morphometric analysis consisted of the measurement of internal and external perimeters of the tunica intima-media and then the inference of the internal and external diameters and media thickness. The values were averaged over 3 measurements for each vessel.

### 4.7. Vascular Reactivity

Experiments were performed on freshly collected rat aortas as previously described [63], using contractility equipment and software (EMKA Technologies, Paris, France). Arterial segments were mounted between two stainless steel hooks placed in a conventional vertical organ bath chamber filled with 5 mL of PSS, maintained at 37 °C and continuously bubbled with O_2_. Changes in isometric tension were measured using an IT1-25 force transducer and an IOX computerized system. Each arterial segment was subjected to a 60 min equilibration period at a basal resting tension of 2 g. Arterial contractility was assessed with phenylephrine (Phe, 10 µM). In some arterial rings, the endothelium was gently removed. The functionality or the absence of the endothelium was tested by the respective ability of acetylcholine (Ach, 1 µM) to induce or not induce relaxation in Phe-contracted rings. After several washouts and a 20–30 min period of stabilization, contraction was evaluated by cumulative increases in the concentration of the agonist phenylephrine (Phe, 0.01–100 µM range, only in the presence of endothelium) or the depolarizing agent KCl (1–80 mM) and a maximally active concentration of a thromboxane A2 agonist, U46619 (1 µM). Endothelial function was assessed by studying the relaxing effects of cumulative increases in acetylcholine concentration (Ach, 1 nM to 10 µM) on arteries contracted with a submaximally active concentration of Phe (1 µM). The effects of the nitric oxide (NO)-synthase inhibitor N^ω^-nitro-L-arginine methyl ester (L-NAME, 10 µM) and the cyclooxygenase inhibitor indomethacin (10 µM) on the relaxing effect of Ach were evaluated. Inhibitors were added for a 15 min period of incubation before PE addition. Endothelium-independent relaxations in response to sodium nitroprusside (SNP, 1 nM–1 µM) were studied in endothelium-denuded rings previously contracted with Phe (10 µM). Each protocol was performed in triplicate for each aorta.

### 4.8. Data Analysis

Three groups were analyzed with the following numbers of individuals: Ctrl (*n* = 20), ELA-LPS (*n* = 20) and ELA-LPS-BB (*n* = 12). Data were expressed as mean ± s.e.m and analyzed using GraphPad Prism (V6.05, RRID:SCR_002798) with one-way ANOVA followed by Tukey’s post hoc test to compare all groups. Concentration–response curves were fitted with non-linear regressions, and statistical differences were assessed using two-way ANOVA followed by Bonferroni’s post hoc test to compare all groups. EC_50_ values referred to the concentration of the drug that induced a response halfway between the baseline and maximum. They were determined for each ring and for each agonist or antagonist and then averaged. The statistical analysis shown compared Ctrl (*) or ELA-LPS (§) to other groups. A *p* value lower than 0.05 was considered significant.

## 5. Conclusions

The presence of both HFpEF and elevated BP in the ELA-LPS COPD-emphysema rat model, despite the absence of systemic endothelial dysfunction, suggests that endothelial dysfunction is not the primary trigger for cardiovascular comorbidities, particularly hypertension, in COPD. In this model without cigarette smoke exposure, the functional and structural remodeling can be interpreted as a secondary adaptation to the increased BP. Moreover, the preventing effects of bisoprolol indicate a significant role of HFpEF in the elevation of BP. The mechanistic relationship between HFpEF, cardiac sympathetic activation and blood pressure requires further investigations in this COPD-emphysema model, as well as in COPD patients.

## Figures and Tables

**Figure 1 ijms-24-12609-f001:**
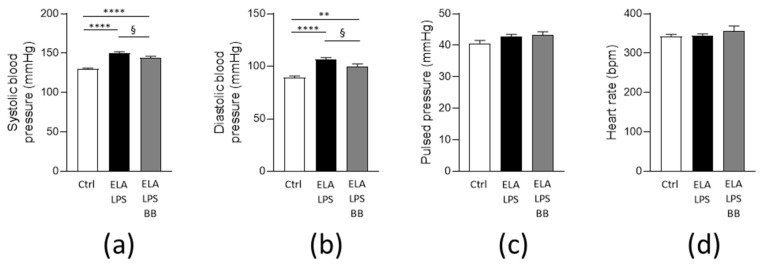
Cardiac characterization of in vivo arterial pressure in vigil animals. Bar graphs represent systolic (**a**), diastolic (**b**) and pulsed (**c**) arterial pressures and heart rate (**d**) measured by tail-cuff for all animals in each group. Data were expressed as mean ± s.e.m for Ctrl (*n* = 20), ELA-LPS (*n* = 20) and ELA-LPS-BB (*n* = 12). ** *p* < 0.01, **** *p* < 0.0001 for comparison vs. Ctrl; ^§^
*p* < 0.05 for comparison vs. ELA-LPS.

**Figure 2 ijms-24-12609-f002:**
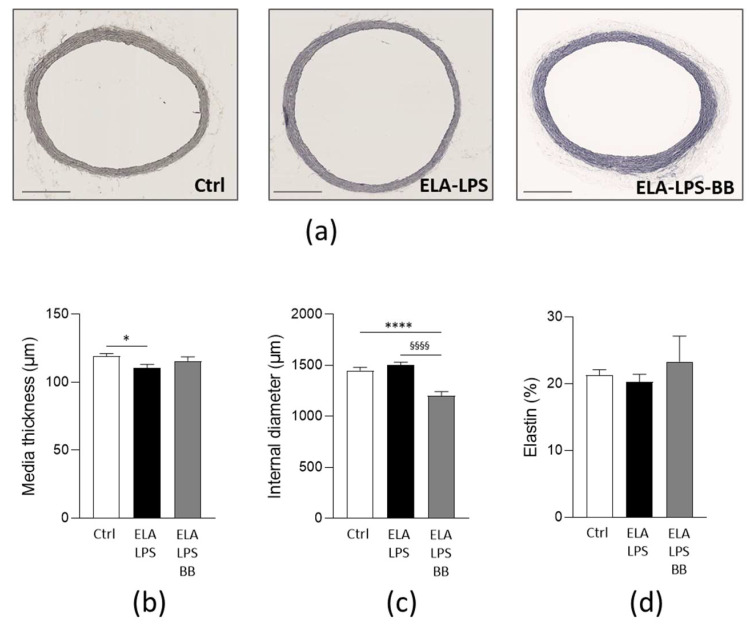
Cardiac morphological characterization of aorta. Bright-field images illustrate representative histological Van Gieson staining of rat aorta sections and illustrate elastin fiber structure and media thickness (**a**). Scale bar = 500 µm. Bar graphs summarize media thickness (**b**), internal diameter (**c**) and elastin ratio (**d**) for Ctrl (*n* = 13), ELA-LPS (*n* = 13) and ELA-LPS-BB (*n* = 10). Data are presented as mean ± s.e.m. * *p* < 0.05, **** *p* < 0.0001 for comparison vs. Ctrl; ^§§§§^
*p* < 0.001 for comparison vs. ELA-LPS.

**Figure 3 ijms-24-12609-f003:**
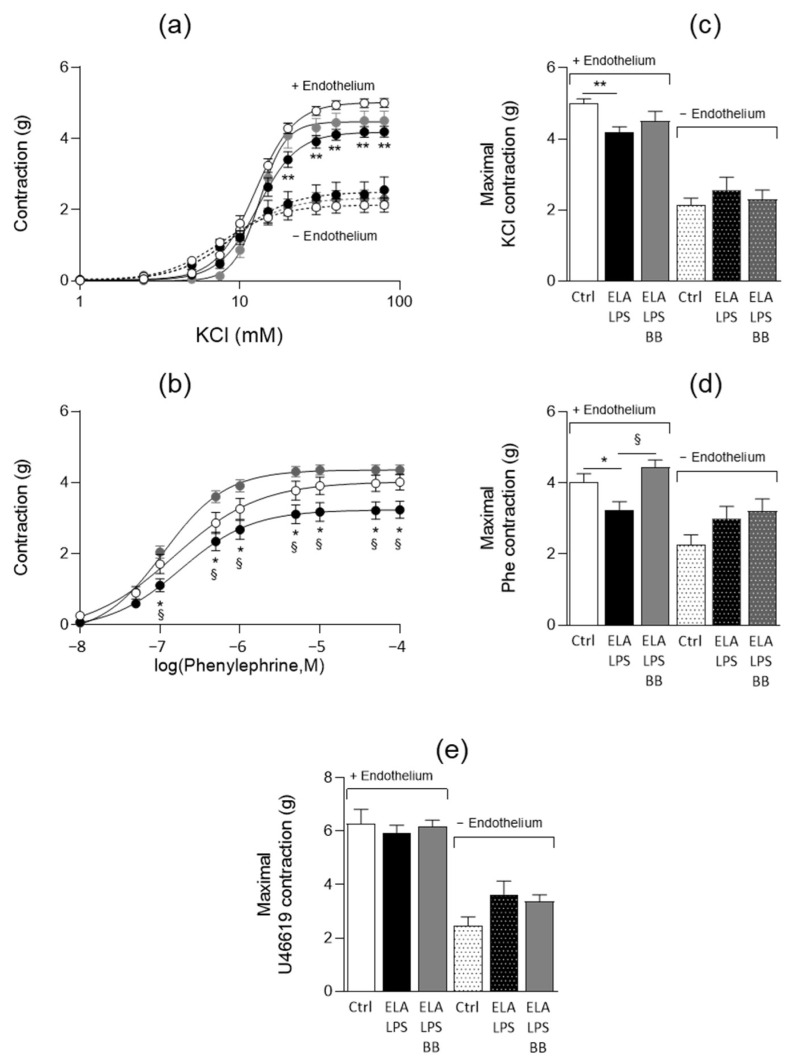
Contractile responses of aorta. Curves summarize cumulative dose-responses to KCl (**a**) and phenylephrine (**b**) in aortic rings from Ctrl (open circle), ELA-LPS (black circle) and ELA-LPS-BB (grey circle) with (line) and without endothelium (dotted line) for KCl. Bar graphs present the maximal contractions induced by KCl (**c**), phenylephrine (**d**) and the thromboxane A2 agonist U46619 (**e**) with and without endothelium in all experimental groups. Data represent mean ± s.e.m for Ctrl (*n* = 20), ELA-LPS (*n* = 20) and ELA-LPS-BB (*n* = 12). ** *p* < 0.01, * *p* < 0.05 for comparison vs. Ctrl; ^§^
*p* < 0.05 for comparison vs. ELA-LPS.

**Figure 4 ijms-24-12609-f004:**
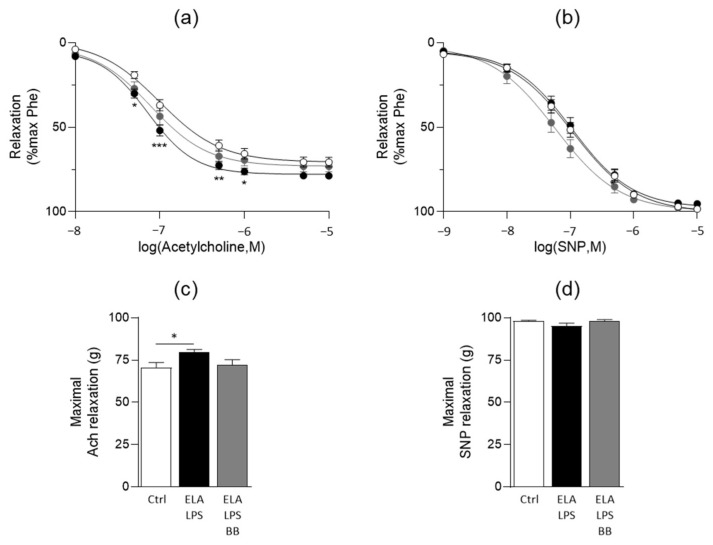
Vasorelaxant responses of rat aorta. Curves summarize cumulative dose-responses to acetylcholine (**a**) and sodium nitroprusside (SNP) (**b**) in aortic rings from Ctrl (white symbol), ELA-LPS (black symbol) and ELA-LPS-BB (grey symbol). Bar graphs present the maximal relaxations induced by acetylcholine (**c**) and SNP (**d**) in all experimental groups. Data represent mean ± s.e.m for Ctrl (*n* = 20), ELA-LPS (*n* = 20) and ELA-LPS-BB (*n* = 12). *** *p* < 0.001, ** *p* < 0.01, * *p* < 0.05 for comparison vs. Ctrl.

**Figure 5 ijms-24-12609-f005:**
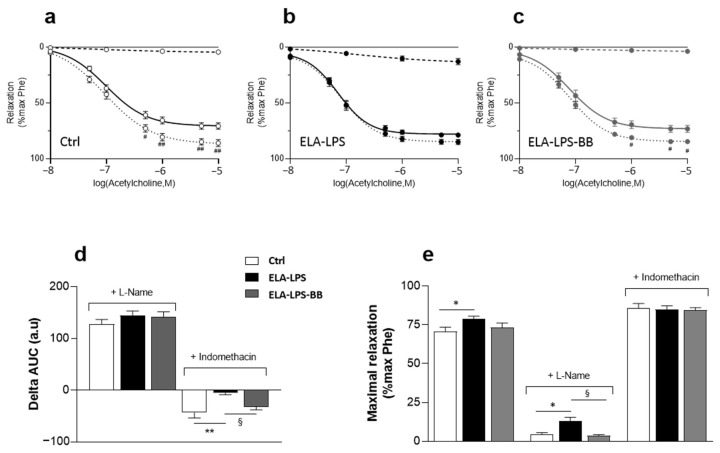
Effect of L-Name and indomethacin on acetylcholine-induced vasorelaxant response. Curves summarize cumulative dose-responses to acetylcholine in the absence (plain line) and in the presence of L-name (hashed line) and indomethacin (dotted line) in aortic rings from Ctrl (**a**), ELA-LPS (**b**) and ELA-LPS-BB (**c**). Bar graphs present the difference in the area under curve between relaxation with and without L-name or indomethacin for each group (**d**) and maximal relaxation for the different conditions (**e**). Data are expressed as mean (*n* = 20 for Ctrl and ELA-LPS, and *n* = 12 for ELA-LPS-BB). ^#^
*p* < 0.05, ^##^
*p* < 0.01 for comparison of dose responses with and without indomethacin for each group; * *p* < 0.05, ** *p* < 0.01 for comparison vs. Ctrl; ^§^
*p* < 0.05 for comparison vs. ELA-LPS.

**Table 1 ijms-24-12609-t001:** Cardiac functional and physiological parameters.

	Ctrl (*n* = 20)	ELA-LPS (*n* = 20)	ELA-LPS-BB (*n* = 12)
Weight (g)	475 ± 7	466 ± 10	470 ± 13
V′O_2_ max (mL/kg/min)	66 ± 1	60 ± 1 ***	64 ± 2
Heart rate (bpm)	357 ± 4	354 ± 5	357 ± 15
LVEF (%)	63.4 ± 1.5	65.6 ± 2.1	66.1 ± 2.4
Cardiac output (mL/min)	125 ± 32	114 ± 26	112 ± 16
E/e′	31.5 ± 2.3	42.2 ± 3.9 *^§^	27.8 ± 4.8

Data representing mean ± s.e.m were compared using one-way ANOVA followed by Tukey’s post hoc test. * *p* < 0.05, *** *p* < 0.001 for comparison vs. Ctrl; ^§^ *p* < 0.05 for comparison vs. ELA-LPS. V′O_2_ max: maximal oxygen uptake; LVEF: left ventricle ejection fraction.

**Table 2 ijms-24-12609-t002:** Aorta sensitivities to vasoactive agonists and antagonist as reflected by EC_50_ values.

	Ctrl (*n* = 20)	ELA-LPS (*n* = 20)	ELA-LPS-BB (*n* = 12)
KCl (mM)	12.7 ± 0.6	13.6 ± 1.1	13.56 ± 0.7
Phe (nM)	147 ± 60	180 ± 44	111 ± 11 ^§^
SNP (nM)	22.2 ± 4.4	18.7 ± 3.7	8.6 ± 2.2
Ach (nM)	122 ± 12	74 ± 5 *	94 ± 18
Ach + L-Name (nM)	85	153 *	82 ^§^
Ach + indo	117 ± 18	89 ± 12	84 ± 11

Data representing mean ±s.e.m were compared using one-way ANOVA followed by Tukey’s post hoc test. * *p* < 0.05, for comparison vs. Ctrl; ^§^
*p* < 0.05 for comparison vs. ELA-LPS. Phe: phenylephrine; SNP: sodium nitroprusside; Ach: acetylcholine; L-Name: N^ω^-nitro-L-arginine methyl ester.

## Data Availability

The data presented in this study are available on reasonable request from the corresponding author.

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
