# Peer review of "Elevated Blood Pressure Occurs without Endothelial Dysfunction in a Rat Model of Pulmonary Emphysema"

_ijms, 2023, doi:10.3390/ijms241612609_

Round 1

Reviewer 1 Report

In the present article, the authors addressed elevated blood pressure and systemic vascular impairment in a rat model of COPD with preserved ejection fraction. They tested the cardio selective b1-receptor antagonist, the bispoprolol on such vascular dysfunction.

Since systemic hypertension is a well known comorbidity factor in COPD patients, the present study is interesting and valuable to increase our knowledge on COPD, a major public health issue.

I am surprised because the model exposed by the authors seems to be very mild although similar model has shown pulmonary hypertension which is supposed to happen in severe models (De oliveira MV et al., Front Physiol, 2019, doi: 10.3389/fphys.2019.00664). Such discrepancy should be discussed. In the article from De Oliveira et al., arterial blood gases show strong significant decrease of PaO2/FiO2. The authors should check arterial blood gases in the present article for comparison.

Altogether the vascular effects observed are very mild (usually less than 20%) and the effect of bisoprolol is low or not significant. This means that sympathetic activation may not be very important in this model. This should be discussed.

Limits of the studies should be discussed : it would be interesting to test what happens in a more severe COPD model. What happens when ELA treatment is longer ?

Sometimes, sentences are not exact.

Page 4, lines 158 – 160, in ELA-LPS-BB, the decreased contractility in presence of endothelium was not reduced for KCl (see figure 3c).

Page 4, lines 161 – 163, the aorta sensitivity is significantly modified in the ELA-LPS-BB group for Phe (see table 2) although the sentence indicates that sensitivities were not modified for all groups.

Specific comments

Page 5, figure 3, why the cumulative dose-responses to U46619 is not shown or has not been performed inversely to KCl and Phe ?

Page 5, figure 3, the concentration of U46619 is not given in the legend of the figure.

Page 6, lines 185 to 187, the authors should be cautious about their conclusions because, although the maximal Ach relaxation in the EL-LPS-BB group is not significantly different from the control group it is also not significant from the ELA-LPS group.

Page 6, lines 188 to 189, since the EC50 are not defined, the authors can not state on “sensitivity”.

Page 7, the paragraph related to the figure 5 in the results section is difficult to follow. First of all, # symbol is not defined in the legend of the figure 5. It would also be usefull to add an histogram with the maximal relaxations to Ach in the 3 groups and in the 3 conditions for each group.

The meaning of the sentence page 7, lines 217 to 219 is not clear. The sentence page 6, lines 224 – 226 appears wrong to me: according to figure 5d, Ach maximal relaxations are not identical for all groups in the presence of indomethacin.

There is a problem in the concentrations of the agonists. They are not consistent between the figures and the methods (SNP is 1 nM to 200 mM in the methods and 0.1 nM to 1 µM in the figure 4 ; Ach is 1 nM to 10 mM in the methods and 10 nM to 10 µM in the figure 4).

The discussion is too long and the authors should concentrate the discussion on COPD and what has been already shown in hypertension related to COPD without describing all models of hypertension.

Minor comments

Page 3, line 103, ELA-LPS-BB should be defined when first mentioned.

Page 6, line 182, “endothelial-dependant” should be added to the sentence in front of “relaxant”

Page 7, line 235, (c) should be replaced by (b).

Author Response

In the present article, the authors addressed elevated blood pressure and systemic vascular impairment in a rat model of COPD with preserved ejection fraction. They tested the cardio selective b1-receptor antagonist, the bisoprolol on such vascular dysfunction.

Since systemic hypertension is a well-known comorbidity factor in COPD patients, the present study is interesting and valuable to increase our knowledge on COPD, a major public health issue.

I am surprised because the model exposed by the authors seems to be very mild although similar model has shown pulmonary hypertension which is supposed to happen in severe models (De oliveira MV et al., Front Physiol, 2019, doi: 10.3389/fphys.2019.00664). Such discrepancy should be discussed. In the article from De Oliveira et al., arterial blood gases show strong significant decrease of PaO2/FiO2. The authors should check arterial blood gases in the present article for comparison.

Authors answer: We first want to thank the reviewer for its general interesting and relevant comments.

For this first point: our results (from this publication as well as from Grillet et al, doi:10.1165/rcmb.2022-0382OC) indeed showed that our ELA-LPS model is a model of COPD with mild severity impairment. Differently from De Oliveira et al., we studied the long-term effects of LPS-induced exacerbation in ELA-induced emphysema rats, with pulmonary and cardiac function assessed 5 weeks after the last LPS instillation. Because De Oliveira et al. explored the pulmonary and cardiac functions 24 hours after the LPS instillation in ELA-induced emphysema rats, lung ultrastructural analyses revealed evidence of increased alveolar collapse, detachment of alveolar type II epithelial cells, endothelial-cell damage and interstitial oedema, infiltration of neutrophils and macrophages in lung tissue. These acute pulmonary impairment was not observed in our study’s animals. This is an important point that could explain the differences observed between the two studies and may reveal different adaptive remodelling.

As information, we performed arterial blood gases analysis that revealed no difference in PO2 between Ctrl and ELA-LPS groups (85+/-10 vs 83+/-12, p = 0.53), in PCO2 (41 +/- 8 vs 43 +/- 11, p=075) and HCO3- (27+/-3 vs 29+/-4, p = 0.99). These results are consistent with a mild severity of lung impairment despite the presence of emphysema. De Oliveira et al. showed significantly decreased PaO2/FiO2 values in LPS and in ELA-LPS but not in ELA suggesting that these modifications are related to LPS instillation and thus to the acute phase of exacerbation.

Altogether the vascular effects observed are very mild (usually less than 20%) and the effect of bisoprolol is low or not significant. This means that sympathetic activation may not be very important in this model. This should be discussed.

Author’s answer: The authors thank this reviewer for this relevant comment. Even if the mean arterial pressure increase is low, it was close to 17,9% (p<0.001). The bisoprolol-induced decrease in mean arterial pressure was close to 5% (p<0.05) and represented a third of the increase. Even if the pressure variations are small because the model is of slight severity, the effect of bisoprolol is consistent. However, bisoprolol was able to fully prevent the cardiac diastolic dysfunction, with the prevention of PKA activation pathway (Grillet et al). Moreover, we observed that bisoprolol treatment was able to fully prevent the increased phenylephrine-induced response and acetylcholin-induced relaxation. We agree that sympathetic activation may partly explain blood pressure increase, but our results show that it may explain a significant part of the vascular remodelling and functional adaptation in the large vessels. This partial effect on blood pressure was evocated (lines 352-356).

Limits of the studies should be discussed: it would be interesting to test what happens in a more severe COPD model. What happens when ELA treatment is longer ?

Author’s answer: Thank you for this comment.

This is an interesting point. While our main objective was to characterize the vascular remodeling in a valid model where an increase in blood pressure and an HFpEF have already been described, the adaptive vascular remodeling would in a more severe model of COPD should constitute an issue for further studies. We also have increased elastase dose (up to 10 UI) and we did not observe significantly more severe emphysema, diastolic dysfunction or increased blood pressure (Wynands et al, 2023, doi: 10.1183/13993003.congress-2022.2866). We did not perform a longer ELA treatment because a 4-week treatment is quite long and corresponds to the longest treatments usually performed for COPD rat model (De Oliveira et al, 2016, doi: 10.3389/fphys.2016.00457).

A “study limitations” paragraph has been added in the Discussion section page 10 (lines 357-362). “Some limitations of this study would be first that the animals were not exposed to cigarette smoke, which is a major, although not systematic, factor in COPD. However, no changes in vascular reactivity of the aorta were previously reported in a guinea pig model of cigarette smoke-induced emphysema [35], suggesting that our findings are not irrelevant in the context of COPD. Second, our animal model is a mild severity model of COPD as revealed by the mild cardiovascular impairments. This could limit the assessment of pathological mechanisms involved in the vascular impairment in COPD.”

Sometimes, sentences are not exact:

Page 4, lines 158 – 160, in ELA-LPS-BB, the decreased contractility in presence of endothelium was not reduced for KCl (see figure 3c).

Author’s answer: Thank you for this careful revision of the text. Indeed, KCl contraction was still decreased in ELA-LPS-BB. We modified the text lines 158-160.  Now it is: “In ELA-LPS-BB, the decreased contractility in presence of endothelium was reduced for KCl and was fully prevented for Phe (p=0.01 vs ELA-LPS) while it was still observed for KCl”

Page 4, lines 161 – 163, the aorta sensitivity is significantly modified in the ELA-LPS-BB group for Phe (see table 2) although the sentence indicates that sensitivities were not modified for all groups.

Author’s answer: We agree with that comment and changed the text according to it. Now we wrote: “The aorta sensitivities to KCl and Phe were not modified between Ctrl and ELA-LPS as reflected by identical EC50 values (Table 2). On the other hand, we observed a decreased EC50 value for Phe in ELA-LPS-BB compared to ELA-LPS that traduced increased Phe sensitivity in ELA-LPS-BB and could reveal adrenergic activation in ELA-LPS.”

Specific comments

Page 5, figure 3, why the cumulative dose-responses to U46619 is not shown or has not been performed inversely to KCl and Phe ?

Dose-responses to U46619 has not been performed inversely to KCl and Phe. Given the duration of the protocol for dose-responses and the limited number of aortic rings for each condition, we prefer to concentrate on KCl and Phe for dose-response as no difference was first observed for U46619. U46619 was added in each organ bath at the end of the experiment.

We first wrote that a maximally active concentration of U46619 was used (line 508).

Page 5, figure 3, the concentration of U46619 is not given in the legend of the figure. Ok, done

Page 6, lines 185 to 187, the authors should be cautious about their conclusions because, although the maximal Ach relaxation in the EL-LPS-BB group is not significantly different from the control group it is also not significant from the ELA-LPS group.

We agree and try to moderate our conclusion. We modified the sentences which are now: “BB treatment partially modified Ach relaxation. Indeed, Emax value was identical to both Ctrl (p=0.94) and ELA-LPS (p=0.2) suggesting an effect of BB treatment with a slight tendency to decrease Ach-induced relaxation otherwise increased in ELA-LPS.”

Page 6, lines 188 to 189, since the EC50 are not defined, the authors cannot state on “sensitivity”.

We have now defined EC50 in the methods section lines 525-528.

Page 7, the paragraph related to the figure 5 in the results section is difficult to follow. First of all, # symbol is not defined in the legend of the figure 5. It would also be useful to add an histogram with the maximal relaxations to Ach in the 3 groups and in the 3 conditions for each group.

Thank you for this suggestion, we have now added a supplemental graph as figure 5e showing maximal relaxations for each condition. Concerning # symbol, the legend has been corrected.

The meaning of the sentence page 7, lines 217 to 219 is not clear. The sentence page 6, lines 224 – 226 appears wrong to me: according to figure 5d, Ach maximal relaxations are not identical for all groups in the presence of indomethacin.

In the new figure 5e, it is now clear that in the presence of indomethacin, ach maximal relaxation are identical in all groups. Thus, indomethacin abolished the difference between groups for ach-induced relaxation.

There is a problem in the concentrations of the agonists. They are not consistent between the figures and the methods (SNP is 1 nM to 200 mM in the methods and 0.1 nM to 1 µM in the figure 4 ; Ach is 1 nM to 10 mM in the methods and 10 nM to 10 µM in the figure 4).

Thank you for that comment. Indeed, there was typing errors in the methods. We have corrected it.

The discussion is too long and the authors should concentrate the discussion on COPD and what has been already shown in hypertension related to COPD without describing all models of hypertension.

The reviewer is right and the discussion has been shortened

Minor comments

Page 3, line 103, ELA-LPS-BB should be defined when first mentioned. Ok, done

Page 6, line 182, “endothelial-dependant” should be added to the sentence in front of “relaxant”; Ok done

Page 7, line 235, (c) should be replaced by (b). Ok, done

Reviewer 2 Report

The manuscript's authors, "Elevated blood pressure occurs without endothelial dysfunction in a rat model of pulmonary emphysema,described the presence of HFpEF and elevated BP using a COPD-Emphysema rat model (ELA-LPS)

This is a well-written manuscript, the work is well presented, and the results support the conclusions and discussion where the authors claimed that despite the absence of systemic endothelial dysfunction, suggests that endothelial dysfunction is not the primary trigger for cardiovascular comorbidities, particularly hypertension, in COPD, which is true for the animal model tested. 

However, this animal model lacks cigarette smoke exposure, the major risk factor for COPD development, which also can induce endothelial dysfunction and cardiovascular comorbidities. This important aspect of the limitation of the study needs to be extensible described, and included in the discussion and conclusion before the manuscript is accepted.

Author Response

The manuscript's authors, "Elevated blood pressure occurs without endothelial dysfunction in a rat model of pulmonary emphysema"  described the presence of HFpEF and elevated BP using a COPD-Emphysema rat model (ELA-LPS).

This is a well-written manuscript, the work is well presented, and the results support the conclusions and discussion where the authors claimed that despite the absence of systemic endothelial dysfunction, suggests that endothelial dysfunction is not the primary trigger for cardiovascular comorbidities, particularly hypertension, in COPD, which is true for the animal model tested.

However, this animal model lacks cigarette smoke exposure, the major risk factor for COPD development, which also can induce endothelial dysfunction and cardiovascular comorbidities. This important aspect of the limitation of the study needs to be extensible described, and included in the discussion and conclusion before the manuscript is accepted.

Author’s answer: We first thank the reviewer for its kind comments.

In animal models, COPD can be induced by different types of exposure. If cigarette smoke is, indeed, one of the most widespread, cardiovascular comorbidities in COPD are not only linked to cigarette smoke. Eickhoff et al. (ref 58, doi:10.1164/rccm.200709-1412OC) assumed that factors other than smoking exert greater influences on systemic vascular function in patients with COPD. Thus, animal models exposed to cigarette smoke seemed not to be the best models for cardiovascular investigations in COPD (Ferrer 2009, Khedoe 2016). Additionally, as we had previously observed HFpEF and elevated blood pressure in this ELA-LPS model of pulmonary emphysema (ref 25, Grillet et al, doi:10.1165/rcmb.2022-0382OC), characterizing vascular function or dysfunction in this model seemed relevant for us.

However, we have added in the discussion section the potential role of cigarette smoke exposure which has not been addressed. As “study limitations”, the following paragraph has been added in the Discussion section page 10 (lines 358-364): “Some limitations of this study would be first that the animals were not exposed to cigarette smoke, which is a major, although not systematic, factor in COPD. However, no changes in vascular reactivity of the aorta were previously reported in a guinea pig model of cigarette smoke-induced emphysema [35], suggesting that our findings are not irrelevant in the context of COPD. Second, our animal model is a mild severity model of COPD as revealed by the mild cardiovascular impairments. This could limit the assessment of pathological mechanisms involved in the vascular impairment in COPD.”

In addition, in the Conclusion section, page 13, we have specified the lack of cigarette smoke exposure in our animal model of COPD (line 536): “In this model without cigarette smoke exposure, the functional and structural remodeling observed in this model can be interpreted as a secondary adaptation to the in-creased BP”